# Immunogenicity and Safety of BNT162b2 Homologous Booster Vaccination in People Living with HIV under Effective cART

**DOI:** 10.3390/vaccines10081243

**Published:** 2022-08-03

**Authors:** Laura Gianserra, Maria Gabriella Donà, Eugenia Giuliani, Christof Stingone, Martina Pontone, Anna Rita Buonomini, Massimo Giuliani, Fulvia Pimpinelli, Aldo Morrone, Alessandra Latini

**Affiliations:** 1STI/HIV Unit, San Gallicano Dermatological Institute IRCCS, 00144 Rome, Italy; laura.gianserra@ifo.it (L.G.); christof.stingone@ifo.it (C.S.); annarita.buonomini@ifo.it (A.R.B.); massimo.giuliani@ifo.it (M.G.); alessandra.latini@ifo.it (A.L.); 2Scientific Direction, San Gallicano Dermatological Institute IRCCS, 00144 Rome, Italy; eugenia.giuliani@ifo.it (E.G.); aldo.morrone@ifo.it (A.M.); 3Microbiology and Clinical Pathology, San Gallicano Dermatological Institute IRCCS, 00144 Rome, Italy; martina.pontone@ifo.it (M.P.); fulvia.pimpinelli@ifo.it (F.P.)

**Keywords:** PLWH, booster, vaccine, COVID-19, SARS-CoV-2, humoral response, SNHL, immunogenicity, safety

## Abstract

Data on COVID-19 boosting vaccination in people living with HIV (PLWH) are scant. We investigated the immunogenicity and safety of the BNT162b2 homologous boosting vaccination. Anti-SARS-CoV-2 spike antibodies (LIAISON^®^ SARS-CoV-2 S1/S2 IgG test, DiaSorin^®^), CD4+, CD8+ and viraemia were monitored at T0 (pre-vaccination), T1 (4 weeks after the second dose), T2 (pre-booster) and T3 (4 weeks after the booster dose). Humoral responses were evaluated according to sex, age, BMI, nadir and baseline CD4+ counts, as well as type of cART regimen. Forty-two subjects were included: the median age was 53 years (IQR: 48–61); the median time since HIV was 12.4 years (IQR: 6.5–18.3); the median nadir and baseline CD4+ counts were 165 (IQR: 104–291) and 687 cells/mm^3^ (IQR: 488–929), respectively. The booster dose was administered at a median of 5.5 months after the second dose. Median anti-SARS-CoV-2 IgG concentration had significantly decreased at T2 compared to T1 (107 vs. 377, *p* < 0.0001). Antibody levels elicited by the booster dose (median: 1580 AU/mL) were significantly higher compared with those of all the other time points (*p* < 0.0001). None of the investigated variables significantly affected antibody response induced by the booster dose. Local and systemic side-effects were referred by 23.8% and 14.3% of the subjects, respectively. One patient developed sensorineural hearing loss (SNHL) 24 h after boosting. He recovered auditory function upon endothympanic administration of corticosteroids. The BNT162b2 boosting vaccination in PLWH is safe and greatly increased the immune response with respect to the primary vaccination.

## 1. Introduction

As of 20 June 2022, almost 40 million people have been vaccinated with the booster dose in Italy (approximately 80% of eligible people) [1]. The Italian Vaccination Plan included in the priority category for anti-COVID-19 vaccination HIV-infected patients with acquired immune deficiency syndrome (AIDS) or CD4+ count <200 cells/mm^3^ [2]. Indeed, several HIV-related factors (immune dysfunction, communicable and noncommunicable comorbidities) might increase risk for severe or fatal disease of people living with HIV (PLWH) infected with SARS-CoV-2. An increased risk of mortality for these subjects has been reported in different meta-analyses [3,4,5]. In addition, a large multicountry analysis has recently shown that HIV infection is independently associated with increased odds of severe/critical COVID-19 and increased likelihood of in-hospital mortality [6]. For all the above-mentioned reasons, anti-COVID-19 vaccination of PLWH is crucial. Unfortunately, waning immunity [7] and reduced vaccine efficacy/effectiveness against infection and symptomatic disease by 6 months after the two-dose primary vaccination have been reported [8]. In addition, because of the emergence and circulation of new variants, a COVID-19 booster dose has been recommended. The booster dose increased the magnitude of the immune response in a pilot trial [9] and provided 95.3% efficacy against COVID-19 in a larger trial [10]. Yet, data on the safety and immunogenicity of boosting vaccination in PLWH are very scant. Tan et al. evidenced significant increases in humoral immunity following the third dose with inactivated vaccine [11]. However, a lower level of neutralizing antibodies was detected compared to healthy individuals. No serious adverse reactions after the booster dose were reported in this study. Another study that used a primary mRNA vaccination for most of the participants, followed by a booster dose with an mRNA-based vaccine, reported that antibody levels after the third dose substantially exceeded those mounted after the primary series [12]. We have previously shown the immunogenicity and safety of primary vaccination (two doses) with the SARS-CoV-2 mRNA BNT162b2 vaccine [13]. Here, we report updated data about the immunogenicity and safety of the booster dose with the same vaccine.

## 2. Materials and Methods

### 2.1. Study Population

The study population included HIV-infected subjects attending the HIV/AIDS Unit of the San Gallicano Dermatological Institute (Rome, Italy), who had previously completed the two-dose regimen between April and May 2021 and who were willing to receive the booster dose. This was administered between September and October 2021. Information regarding clinical history, anti-retroviral therapy, HIV-1 RNA load (VL) and CD4+ T-cell count at each time point were retrieved from the medical records. Relevant data (height and weight) for the calculation of Body Mass Index (BMI) were collected at enrollment. At the time of the administration of each dose, all participants were interviewed about contacts with symptomatic or asymptomatic SARS-CoV-2 positive individuals; previous or current COVID-19 related symptoms (e.g., cough, cold, fever, shortness of breath, flu-like symptoms, sore throat, diarrhoea, anosmia, ageusia, etc.) and previous testing for SARS-CoV-2. All participants provided a signed informed consent at enrollment. The study was conducted in accordance with the Declaration of Helsinki and approved by the Institutional Ethics Committee (Prot. RS 1463/21).

### 2.2. Immunization Schedule

Following the first two doses of the BNT162b2 vaccine (Pfizer, New York, NY, USA and BioNTech, Mainz, Germany) through intramuscular injection of 30 µg/dose at day 1 and day 22, a third dose of 30 µg of the same vaccine (booster dose) was administered intramuscularly.

### 2.3. Serological Test for SARS-CoV-2 S1/S2 IgG

Blood samples were collected at T0 (pre-vaccination), T1 (post-primary vaccination, i.e., 4 weeks after the second dose), T2 (pre-booster) and T3 (post-booster, i.e., 4 weeks after the booster dose). The LIAISON^®^ SARS-CoV-2 S1/S2 IgG test (DiaSorin^®^, Saluggia, Italy) was used for the detection of IgG antibodies against S1 and S2 subunits of the spike protein. This quantitative chemiluminescent immunoassay (CLIA) was performed on a fully automated LIAISON^®^ XL platform. Following the manufacturer’s instructions, results were classified as (i) nonreactive (<12 Absorbance Unit, AU/mL), (ii) equivocal (12–14.9 AU/mL) and (iii) reactive (≥15 AU/mL).

### 2.4. Safety

Patients were asked to report to the clinicians involved in the study both local (redness, swelling or pain at the injection site) and systemic reactions (e.g., fever, fatigue or headache) that occurred from day 1 to day 15 after the third dose. Patients were also requested to report any other suspected adverse event.

### 2.5. Statistical Analysis

Descriptive statistics were used to summarize the characteristics of the study group. The responder rates were calculated as the percentages of the vaccinated subjects whose serological test results were ≥15 AU/mL. A violin plot with logarithmic transformation was used to show the median antibody concentration, first and third quartiles as well as frequency of data, at the four time points. The Wilcoxon test for paired samples was used to compare median antibody concentrations at different time points. To assess immunogenicity according to subgroups, these were defined by age (cut-off at the median), sex, BMI (normal weight = 18.8–24.9 kg/m^2^, overweight = 25.0–29.9 kg/m^2^, obese ≥ 30 kg/m^2^), nadir CD4+ T-cell count (hereafter CD4+), baseline (T0) CD4+, cART regimen based on integrase strand transfer inhibitor-INSTI (no/yes) and dual therapy (no/yes). The median antibody concentrations of independent groups were compared using the Mann–Whitney or Kruskal–Wallis tests, as appropriate. A *p* value < 0.05 was considered as significant. Statistical analyses were carried out using MedCalc^®^ Statistical Software version 20.111 (MedCalc Software Ltd., Ostend, Belgium; https://www.medcalc.org).

## 3. Results

### 3.1. Study Population

Of the 63 subjects who completed the primary vaccination, 21 (33.3%) were not included in the present study for the following reasons: two were already reactive at T0, three tested positive for SARS-CoV-2 after the second dose, five refused the booster dose, and eleven did not receive the booster dose at our hospital.

The remaining 42 subjects (66.7%) entered the booster group (median age: 53 years, IQR: 48–61; 37 males, 88.1%; median time since HIV diagnosis: 12.4 years, IQR: 6.5–18.3). The median counts of nadir and T0 CD4+ were 165 (IQR: 104–291) and 687 cells/mm3 (IQR: 488–929), respectively. CD8+ T-cell median count at T0 was 701 cells/mm3 (IQR: 614–968). All participants were on cART: 18/42 (42.9%) were receiving a regimen based on integrase strand transfer inhibitors (INSTI), and 15/42 (35.7%) were on dual therapy. Except for one subject, who had 73 copies/mL of HIV RNA, all the participants had undetectable viraemia at T0.

At enrollment (T0), none of the participants referred to have ever had COVID-19-related symptoms or a SARS-CoV-2 positive test. However, we cannot exclude asymptomatic infections or undiagnosed mild disease before testing was implemented (e.g., at the beginning of the pandemic). None of the participants became infected during the study period.

### 3.2. Immunogenicity of BNT162b2 Booster Dose

The booster dose was administered at a median of 5.54 months (IQR: 5.27–5.50) after the second dose. All participants included in the booster group had seroconverted after the primary vaccination (T1). Forty-one of forty-two subjects (97.6%) were still reactive on the day they received the booster dose (T2). All the patients were reactive at T3. Figure 1 shows the anti-SARS-CoV-2 IgG response at the four time points. The median concentration of anti-SARS-CoV-2 IgG significantly increased from T0 to T1 (3.8 vs. 377.0 AU/mL, *p* < 0.0001), but a significant decrease was observed at T2 compared to T1 (107.0 vs. 377.0 AU/mL, *p* < 0.0001). The highest antibody level was reached after the booster dose (median: 1580.0 AU/mL, IQR: 605–2550), with a significant increase compared with T0 (*p* < 0.0001), T1 (*p* < 0.0001) and T2 (*p* < 0.0001).

The Geometric Mean Concentrations (GMCs) of the anti-spike IgG at T0, T1, T2 and T3 were 4.28 (95% CI: 3.74 to 4.89), 411.38 (95% CI: 323.93 to 522.44), 105.07 (95% CI: 77.59 to 142.28) and 1210.65 (95% CI: 886.52 to 1653.28), respectively.

The antibody response after the booster dose did not significantly differ by sex, age, BMI, nadir CD4+ or baseline CD4+ (Table 1). The antibody levels were higher in those treated vs. those not treated with dual therapy (*p* = 0.09) and in those treated vs. those not treated with an INSTI-based regimen (*p* = 0.07), but these differences did not reach statistical significance.

### 3.3. HIV-Related Parameters

Neither CD4+ nor CD8+ T-cell counts were significantly affected by the booster dose. Indeed, CD4+ T-cell median count went from 687 at T0 to 732 cells/mm^3^ at T3 (*p* = 0.11). A nonsignificant increase in CD8+ T-cell median count was observed (701 at T0 vs. 751 cells/mm^3^ at T3, *p* = 0.41). Two female patients (43 and 53 years old) with undetectable viraemia at T0 showed detectable viraemia at T1 (70 and 47 copies/mL, respectively). At T2, viral load was undetectable for both patients, while another blip was observed for the 43-year-old woman after the booster dose (195 copies/mL).

### 3.4. Safety of BNT162b2 Booster Dose

During the 15 days after the booster dose, 10 out of 42 patients (23.8%) reported local side effects, i.e., mild pain at the injection site that subsided within 48 h. Systemic side effects were referred by 6/42 subjects (14.3%). In detail, headache and fever (<24 h) were reported in two (4.8%) and three cases (7.1%), respectively; one subject reported mild arthromyalgia (2.4%). Additionally, a 58-year-old man developed hearing loss 24 h after the booster dose, but no tinnitus, vertigo or dizziness were reported. This patient referred a previous episode of sensorineural hearing loss (SNHL), with a mild function loss at the right ear. The audiometric curve after boosting indicated a condition of bilateral SNHL. The patient was treated with endothympanic administration of corticosteroid and recovered auditory function and audiometric characteristics as prior to vaccination in about 1 month.

## 4. Discussion

We describe the results of a prospective observational study investigating the safety and immunogenicity of the BNT162b2 booster dose in PLWH. In our study population, antibody levels elicited by the second dose significantly declined in less than 6 months. Before receiving the third dose, a loss of reactivity was even observed in one subject. Consistently with the pilot study conducted with the BNT162b2 vaccine in the general population [9], we observed that the booster dose significantly increased the humoral response of HIV-infected subjects with respect to the primary vaccination. Our findings are in line with those by others in PLWH who received an RNA-based booster vaccination, although mostly with the mRNA-1273 vaccine [12]. The evidence of a greater humoral immunity elicited by the booster dose compared with primary vaccination alone was also confirmed by another study in PLWH that, however, used an inactivated COVID-19 booster vaccination [11]. In line with our previous observations on the HIV-infected subjects receiving the primary vaccination [13], we observed that neither age, sex, BMI, nadir CD4+, baseline CD4+ nor cART regimen significantly affected the humoral response after the third dose. These findings are consistent with what was reported by Jedicke et al. with respect to nadir and current CD4+ [14] and by Lapointe et al. with respect to nadir CD4+ [12]. Differently from what emerged from our study, the ORCHESTRA project, which investigated the humoral immune response to the booster dose with our same vaccine in PLWH, demonstrated that a CD4+ count <200 cell/mm^3^ at baseline was significantly associated with a lower antibody response in comparison with a CD4+ count >500 cell/mm^3^ [15]. No changes in CD4+ or CD8+ counts were observed after the administration of the third dose, as we observed after the primary vaccination [13]. A significant decline of CD4+ count has been described by others, but this variation was observed only after the first and second doses of BNT162b2 [16].

After booster vaccination, we observed a blip in one female patient who had already blipped after the second dose. This patient had a history of suboptimal therapy adherence, which most likely explains our observations. However, a transient positive viraemia has been reported by others after the primary vaccination with mRNA-based COVID-19 vaccines [16,17].

Our study showed that the BNT162b2 booster dose is safe in HIV-infected subjects. Indeed, the adverse effects were mild and similar to those we observed after the primary vaccination [13]. The percentages of participants who reported local and systemic events are in accordance with those described in other populations who received a booster dose with BNT162b2 [10,18]. Regarding SNHL, despite the initial hypothesis of SNHL being a potential side effect of vaccination against SARS-CoV-2, a US study concluded that there is no association between SNHL and the administration of mRNA-based SARS-CoV-2 vaccines [19].

Our study has a few limitations. Firstly, our study group was rather small. We cannot exclude that significant differences in the humoral response according to the investigated variables might have emerged with a larger sample size. Moreover, the study only included individuals under cART with good immune recovery. Therefore, our findings are not generalizable to other populations of PLWH. In addition, we lacked appropriately matched healthy controls for humoral response comparison analysis. Finally, we only evaluated humoral response.

## 5. Conclusions

In conclusion, our study showed that the BNT162b2 homologous booster regimen is immunogenic and safe in HIV-infected subjects who are receiving cART and have good immune recovery. Notably, the third dose elicited a significantly higher antibody response compared with the two primary doses. Additionally, the booster dose did not significantly affect CD4+ and CD8+ counts. Further studies are needed to evaluate booster dose immunogenicity in PLWH with different clinical and viro-immunological profiles and to monitor for a possible decline over time of the humoral response elicited by the booster dose. In addition, given the fact that the neutralization of the currently circulating variants (BA.4 and BA.5) is reduced even in people immunized with triple doses of the BNT162b2 vaccine [20], it cannot be excluded that enhanced vaccination strategies should be considered for PLWH.

## Figures and Tables

**Figure 1 vaccines-10-01243-f001:**
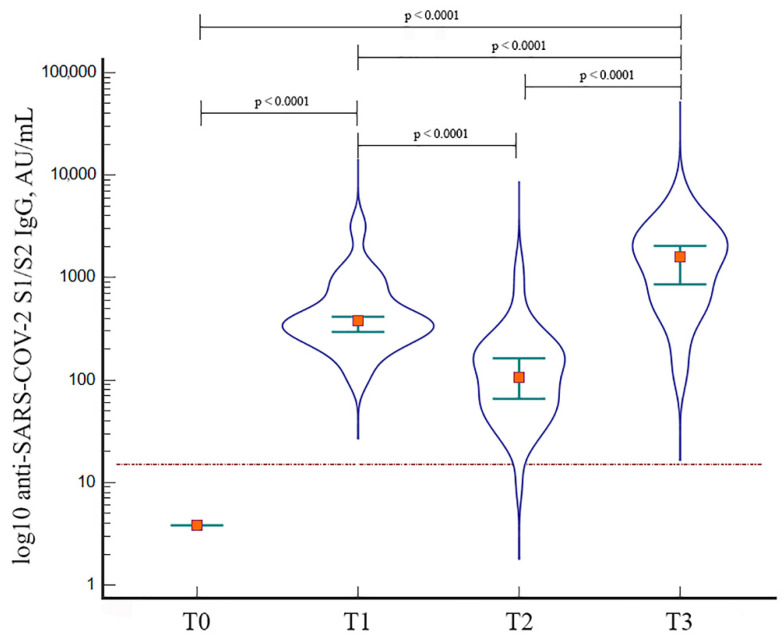
Violin plot showing the median concentration (orange box) of anti-SARS-CoV-2 S1/S2 IgG at the four time points: T0 (pre-vaccination), T1 (4 weeks after the second dose), T2 (pre-booster) and T3 (4 weeks after the booster dose). Green bars show the IQR (25–75 percentiles). The red line indicates the cut-off for reactivity (15 AU/mL).

**Table 1 vaccines-10-01243-t001:** Median anti-SARS-CoV-2 S1/S2 IgG measured 4 weeks after BNT162b2 booster dose in 42 HIV-infected individuals according to demographical and clinical variables.

Variable	N	Median Anti-S1/S2 IgG (AU/mL)	25th–75th Percentile	*p*-Value
Sex				0.74
men	37	1530	636 to 2543	
women	5	1670	474 to 4160	
Age, years				0.67
up to 53	22	1580	659 to 2590	
>53	20	1590	519 to 2545	
BMI				0.74
normal weight	15	1770	860 to 2183	
overweight	19	1370	495 to 2763	
obese	8	963	519 to 2680	
Nadir CD4+, cells/mm^3^				0.81
up to 200	25	1630	600 to 2618	
>200	17	1530	612 to 2298	
Baseline CD4+, cells/mm^3^				0.91
up to 500	12	1375	736 to 2890	
>500	30	1600	586 to 2550	
Dual therapy				0.09
no	27	1270	514 to 2455	
yes	15	1960	1105 to 2763	
INSTI-based regimen				0.07
no	24	967	519 to 2120	
yes	18	1820	1270 to 3240	

## Data Availability

The data presented in this study are available on request from the corresponding author.

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
