# Peer review of "Immunogenicity and Safety of BNT162b2 Homologous Booster Vaccination in People Living with HIV under Effective cART"

_vaccines, 2022, doi:10.3390/vaccines10081243_

Round 1
Reviewer 1 Report
Review of "Immunogenicity and safety of BNT162b2 homologous booster vaccination in people living with HIV"
The title of this manuscript is a bit misleading. Misleading because their study population is a select group of PLWH that are receiving cART and have been previously shown to have good immune responses to the BNT/Pfizer vaccine for COVID-2. The authors point out this limitation in the Discussion but the title implies a more generalizable result. Perhaps an addition to the title at the end, like "under effective cART".
The study design was to investigate IgG antibody responses pre-vaccination, 4 weeks after the second dose of BNT/Pfizer, a pre-booster response (~4-6 months after the second dose, and a post-booster (4 weeks after the booster dose). Their IgG antibody results were consistent with previously observed declines in antibody titers 4-6 months after primary and secondary immunizations with non-HIV infected patients. 4 weeks after the booster, they observed a highly significant increase in IgG antibody titers consistent with results from non-HIV infected patients. It might have been useful to have a cohort of non-HIV infected patients to give some perspective to this study. The patient side effects from this study were more or less in line with what has been previously reported in non-HIV infected patients and the investigators did not observe statistically significant differences by sex, age, BMI, CD4+ nadir or baseline counts, or anti-retroviral therapy regimens. Perhaps a larger patient sample size might have turned up some differences here.
Overall, this small study does contribute some useful information to a population of PLWH that are well controlled in their HIV status. I feel it is important to make this distinction clear in the title of the study as it is already made clear in the manuscript. Not everyone reads the entire article.
Author Response
- The title of this manuscript is a bit misleading. Misleading because their study population is a select group of PLWH that are receiving cART and have been previously shown to have good immune responses to the BNT/Pfizer vaccine for COVID-2. The authors point out this limitation in the Discussion but the title implies a more generalizable result. Perhaps an addition to the title at the end, like "under effective cART".
Authors’ response:
Thank you for your comment. In line with your suggestion, we have now changed the title as follows: “Immunogenicity and safety of BNT162b2 homologous booster vaccination in people living with HIV under effective cART” (page 1).
- The study design was to investigate IgG antibody responses pre-vaccination, 4 weeks after the second dose of BNT/Pfizer, a pre-booster response (~4-6 months after the second dose, and a post-booster (4 weeks after the booster dose). Their IgG antibody results were consistent with previously observed declines in antibody titers 4-6 months after primary and secondary immunizations with non-HIV infected patients. 4 weeks after the booster, they observed a highly significant increase in IgG antibody titers consistent with results from non-HIV infected patients. It might have been useful to have a cohort of non-HIV infected patients to give some perspective to this study. The patient side effects from this study were more or less in line with what has been previously reported in non-HIV infected patients and the investigators did not observe statistically significant differences by sex, age, BMI, CD4+ nadir or baseline counts, or anti-retroviral therapy regimens. Perhaps a larger patient sample size might have turned up some differences here.
Authors’ response:
Thank you for raising this issue. As a limitation of the study, we had already specified that we did not have a comparison group of HIV-uninfected individuals (page 6). As you suggested, we have now added to the study limitations that “significant differences in the humoral response according to the investigated variables might have emerged with a larger sample size” (page 6).
- Overall, this small study does contribute some useful information to a population of PLWH that are well controlled in their HIV status. I feel it is important to make this distinction clear in the title of the study as it is already made clear in the manuscript. Not everyone reads the entire article.
Authors’ response:
On the basis of your consideration, we have now changed the title of the manuscript (page 1).
The article has been revised by a native English speaker.
Reviewer 2 Report
I feel this is a very important article for the scientific community and public health policies. Although, there are few limitations, as mentioned by the authors, there is a real value in disseminating the information presented in the manuscript.
The data is presented in a clear way, using the appropriate material and procedures and with the appropriate statistical analysis
Author Response
-I feel this is a very important article for the scientific community and public health policies. Although, there are few limitations, as mentioned by the authors, there is a real value in disseminating the information presented in the manuscript.
The data is presented in a clear way, using the appropriate material and procedures and with the appropriate statistical analysis.
Authors’ response:
Thank you for the appreciation for our study. The article has been revised by a native English speaker.
Reviewer 3 Report
The study described the time-course of Anti-SARS-CoV-2 spike antibodies in a little cohort of subjects with chronic HIV infection, tested before vaccination, 4 weeks after the second dose and before and after the booster dose. Limited data on humoral response after the third vaccine dose in PLWH are available however, the number of subjects enrolled is low and anti SARS-CoV-2 spike antibodies level was dosed only after an early interval (4 weeks) after the third dose, despite they received the last dose September-October 2021.
The authors reported that no SARS-CoV-2 infection occurred throughout the study period: median anti-SARS-CoV-2 IgG value at T0 was 3.8 AU but a previous asymptomatic infection or as undiagnosed mild disease early before testing (at the beginning of pandemic, for example) can’t be excluded, because of antibody decline overtime.
My questions are:
1-About a third (21 out of 63) of the subjects enrolled were excluded: please describe which are the reason for.
2- The exclusion of a previous SARS-CoV-2 infection was made only on clinical basis? If yes, a specific protocol in the case of clinical suspicion was applied ? Please discuss.
3- Did you find any correlation between age and humoral response to vaccine ?
Author Response
- The study described the time-course of Anti-SARS-CoV-2 spike antibodies in a little cohort of subjects with chronic HIV infection, tested before vaccination, 4 weeks after the second dose and before and after the booster dose. Limited data on humoral response after the third vaccine dose in PLWH are available however, the number of subjects enrolled is low and anti SARS-CoV-2 spike antibodies level was dosed only after an early interval (4 weeks) after the third dose, despite they received the last dose September-October 2021.
The authors reported that no SARS-CoV-2 infection occurred throughout the study period: median anti-SARS-CoV-2 IgG value at T0 was 3.8 AU but a previous asymptomatic infection or as undiagnosed mild disease early before testing (at the beginning of pandemic, for example) can’t be excluded, because of antibody decline overtime.
Authors’ response:
We agree with your comment and we have modified the text accordingly (Results-Study population, page 3).
My questions are:
1-About a third (21 out of 63) of the subjects enrolled were excluded: please describe which are the reason for.
Authors’ response:
Thank you for the suggestion. To clarify, we have now added to the text the reasons why 21 subjects were excluded from the study (Results-Study population, page 3).
2- The exclusion of a previous SARS-CoV-2 infection was made only on clinical basis? If yes, a specific protocol in the case of clinical suspicion was applied ? Please discuss.
Authors’ response:
We have now specified that, at the time of administration of each dose, all participants were interviewed about contacts with symptomatic or asymptomatic SARS-CoV-2 positive individuals; previous or current COVID-19 related symptoms (cough, cold, fever, shortness of breath, flu-like symptoms, sore throat, diarrhoea, anosmia, ageusia,…); previous testing for SARS-CoV-2 (Materials and methods-Study population, page 2).
3- Did you find any correlation between age and humoral response to vaccine?
Authors’ response:
We thank the reviewer for giving us the opportunity to reiterate this point. As described in the Results section “Immunogenicity of BNT162b2 booster dose”, we found no correlation between age and antibody response after the booster dose (page 4).
The article has been revised by a native English speaker.
Round 2
Reviewer 3 Report
The Authors replied to all my questions